# Supraspinatus Muscle Regeneration Following Rotator Cuff Tear: A Study of the Biomarkers Pax7, MyoD, and Myogenin

**DOI:** 10.3390/ijms252111742

**Published:** 2024-11-01

**Authors:** Eva Kildall Hejbøl, Stephanie Wej Andkjær, Julie Dybdal, Marie Klindt, Sören Möller, Kate Lykke Lambertsen, Henrik Daa Schrøder, Lars Henrik Frich

**Affiliations:** 1The Orthopedic Research Unit, Hospital Sønderjylland, Department of Regional Health Research, University of Southern Denmark, 5230 Odense, Denmark; eva.kildall.hejboel@rsyd.dk (E.K.H.);; 2Department of Neurobiology Research, Institute of Molecular Medicine, University of Southern Denmark, 5230 Odense, Denmark; 3Department of Pathology, Odense University Hospital, 5000 Odense, Denmark; 4Open Patient Data Explorative Network, Odense University Hospital, 5000 Odense, Denmark; 5Department of Clinical Research, University of Southern Denmark, 5230 Odense, Denmark; 6Department of Neurology, Odense University Hospital, 5000 Odense, Denmark; 7BRIDGE, Brain Research Inter-Disciplinary Guided Excellence, University of Southern Denmark, 5230 Odense, Denmark; 8Department of Orthopedics, Odense University Hospital, 5000 Odense, Denmark

**Keywords:** muscle regeneration, rotator cuff tear, supraspinatus, myogenic factors, Pax7, MyoD, myogenin

## Abstract

The success of rotator cuff tendon repair relies on both tendon healing and muscle recovery. The objective of this descriptive study was to investigate the regenerative potential of the supraspinatus muscle in rotator cuff tear conditions by quantifying the expression of Pax7, MyoD, and myogenin, basic factors that regulate myogenesis. Muscle biopsies were collected from thirty-three patients aged 34 to 73 years who underwent surgery for a rotator cuff tear affecting the supraspinatus muscle. Among these patients, twenty-seven percent were women, and the age of the lesions ranged from 2 to 72 months post-initial trauma. Biopsies were harvested from the supraspinatus muscle at the end closest to the tendon, and control biopsies were harvested from the ipsilateral deltoid muscle. The densities of immunohistochemically stained Pax7^+^, MyoD^+^, and myogenin^+^ nuclei/mm^2^ were used to estimate the myogenic potential of the muscle. Adjustments were made for patient age and lesion age. We found increased density of MyoD^+^ and myogenin^+^ cells in supraspinatus muscles compared to deltoid muscles (*p* < 0.001 and *p* = 0.003, respectively). Regression analyses that combined the density of positive nuclei with patient age showed a continuous increase in Pax7 with age but also a reduction of MyoD and myogenin in older patients. When combined with lesion age, there was a decline in the density of all myogenic markers after an initial rise. Pax7 density continued to be higher in supraspinatus compared to the deltoid muscle, but the density of MyoD and myogenin terminally dropped to a density lower than in the deltoid. Our findings suggest that the supraspinatus muscle in tear conditions showed signs of initial activation of muscle regeneration. When compared to the unaffected deltoid muscle, an apparent reduction in capacity to progress to full muscle fiber maturity was also demonstrated. This pattern of inhibited myogenesis seemed to increase with both patient age and lesion age. Our results on muscle regenerative capacity indicate that younger patients with rotator cuff tears have better chances of muscle recovery and may benefit from early surgical reconstruction.

## 1. Introduction

The prevalence of rotator cuff (RC) tears is high and increases markedly after 50 years of age [1]. The etiology of RC tears is a combination of tendon degeneration and attritional micro-trauma, as well as direct and indirect traumas against the shoulder [2,3]. In most cases, the supraspinatus tendon is involved, either alone or in combination with tears of one or more tendons [4].

Treatment options are either conservative (physiotherapy, analgesics, and anti-inflammatory drugs) or surgical reattachment of the lesioned tendon. Lack of healing after conservative treatment and tendon re-tear never leads to complete recovery of the rotator cuff muscles [5,6]. One major reason that hampers shoulder function after RC tears is preexisting degenerative changes of the rotator cuff muscles [7,8] consisting of atrophy, fatty infiltration, fibrosis, and inflammation [9,10]. Fatty degeneration and muscle atrophy often appear rapidly after a tear and are prone to persist despite a successful repair [5,11]. This phenomenon is rather unique, as muscular atrophy due to immobilization following a limb cast is often reversible [12,13].

Experimental studies on rabbits and rodents [14,15,16,17] have shown that injection of human satellite cells in RC tears supports muscle regeneration and reduces fibrosis. This suggests a possible therapeutic access to improve clinical outcomes after rotator cuff repair.

Muscle stem cells, the satellite cells, are the primary cells responsible for skeletal muscle regeneration. They are located under the basal lamina of skeletal muscle fibers [18] and express the transcription factor paired box protein 7 (Pax7) [19]. When a muscle is exposed to injury, the satellite cells leave quiescence and reenter the cell cycle [20]. As Pax7 is involved in myogenic cell lineage determination and specification [19,21], Pax7 is expressed in quiescent and early-activated muscle stem cells. When satellite cells are initially activated, myogenic regulatory factors such as myogenic differentiation 1 (MyoD) [22], and, later, myogenin are sequentially expressed and function in the differentiation [23]. Pax7^+^, MyoD^+^, and myogenin^+^ nuclei in a muscle fiber are therefore an expression of the muscle stem cell activity and stage of differentiation.

The aim of this study was to quantify the density of myogenic markers Pax7, MyoD, and myogenin as an expression of regenerative potential in the supraspinatus muscle following RC tear and to compare the results to the healthy ipsilateral deltoid muscle. Moreover, we wanted to study the temporal development of these markers related to patient age and lesion age. This was achieved by harvesting muscle biopsies from the myotendinous zone of the supraspinatus muscle in patients going through surgery for RC tendon tears. From these biopsies, the number of Pax7^+^, MyoD^+^, and myogenin^+^ nuclei/mm^2^ was estimated in immunohistochemically stained tissue sections. Our hypothesis was that a hampered muscle regeneration capacity in the supraspinatus muscle would be reflected in the expression of myogenesis regulating factors.

## 2. Results

### 2.1. Inter-Rater r-Values

The results from the two raters differed, as shown in Appendix A. Inter-rater r-values were as follows for Pax7^+^ (*r* = 0.52), MyoD^+^ (*r* = 0.85), and myogenin^+^ (*r* = 0.87).

### 2.2. Myogenic Capacity and Activity 

The number of MyoD^+^ and myogenin^+^ nuclei/mm^2^ were significantly higher in supraspinatus compared to deltoid muscles (*p* < 0.001 and *p* = 0.003, respectively), while the difference in Pax7 approached statistical significance (*p* = 0.058 (Table 1)).

Lowess analysis combining the nuclear densities with patient age and lesion age showed nonlinear and non-monotonous curves (Figure 1). When looking at the age-associated density patterns (Figure 1A), it appeared that the deltoid muscle generally had a lower density than the supraspinatus muscle, with the exception that Pax7 density in the younger age group up to the late fifties was lower in supraspinatus. When the densities of the myogenic factors were combined with the age of the lesion (Figure 1B), Pax7 was higher in the supraspinatus muscle than in the deltoid, while MyoD and myogenin were initially higher in the supraspinatus but dropped below that of the deltoid in older lesions. Generally, the density patterns varied less in the deltoid than in the supraspinatus.

The age-dependent pattern of Pax7 in the supraspinatus showed a tendency toward an overall increase, while MyoD and myogenin had an initial increase or started high and then decreased.

### 2.3. Density of Myogenic Factors

The density patterns related to the age of the lesion showed an initial rapid increase for Pax7 and MyoD, followed by a slow decrease (Figure 1). Myogenin decreased during the entire period.

In a statistical model adjusted for patient age (continuous) and age of lesion, we analyzed the age-related patterns (Table 2). Our initial decision (see materials and methods) to compare lesions treated within the first three months with older lesions was further supported by the lowess analysis (Figure 1B).

Concerning the relation to patient age, we found a tendency to decrease for myogenin (*p* = 0.064) and a decrease in Pax7 (*p* = 0.007). Comparing lesions less than three months, MyoD showed a tendency to decrease (*p* = 0.078), while myogenin decreased (*p* = 0.007). The only significant finding in the deltoid was an initial higher Pax7 density (*p* = 0.006).

The age of patients and age of lesions in male and female groups were similar (Table 3 in Section 4), and when the groups were compared, no gender difference was found (*p* = 0.878 for patient age and *p* = 0.192 for lesion age).

### 2.4. Presence of Internal Nuclei

In 18 of 33 supraspinatus muscle biopsies, an increased number of internal nuclei were present (Appendix A, Table A1). There was no relation to patient age regarding the presence of internal nuclei (mean 59.5 years in the group with internal nuclei and 61.0 years in patients without internal nuclei). Compared to lesion age, there was a non-significant tendency towards a higher number of patients with internal nuclei in the group with lesions older than three months (*p* = 0.056).

## 3. Discussion

The present study offers a quantitative analysis of the regenerative potential of the supraspinatus muscle after RC tears. The key finding is that, compared to the deltoid muscle, the supraspinatus muscle initially showed increased regenerative capacity, but this declines with both patient age and lesion age, indicating that while the supraspinatus muscle retains regenerative capacity, this capacity progressively diminishes.

This study was based on digital automated quantitation using immunohistochemically stained histological sections with QuPath software version 0.4.3 [24]. Ideally, perfect agreement between the two raters would yield a correlation of 1.0, but our correlations ranged from 0.52 to 0.87, suggesting some variation between raters. This difference may be due to varying threshold values for positive detection, which could create systematic deviations. Other factors, such as how cells are split by shape and decisions to exclude connective tissue and artifacts from the counting area, may also account for differences.

Regarding the preserved regenerative capacity, quantification directly from tissue sections showed that the density of cells expressing myogenic factors was generally higher in the supraspinatus muscle compared to the deltoid muscle, indicating myogenic activation. The presence of internal nuclei in some supraspinatus muscle biopsies could also be interpreted as an activation signal. However, a flow cytometric study by Koide et al. [18] of supraspinatus muscle in torn RCs, with an age distribution similar to ours, found no difference in the number of myogenic cells. On the other hand, Koide et al. reported increased myogenic gene expression in lesioned supraspinatus compared to non-lesioned subscapularis [18]. Their in vitro findings on myoblasts isolated from supraspinatus muscle with damaged tendon, compared to normal subscapularis, showed no differences in the cells’ capacity to proliferate and mature. Meyer et al. [25] further supported this by finding no differences in regenerative capacity between muscle precursors isolated from patients with bursitis and those with partial or full-thickness RC tears, indicating retained myogenic potential.

Fatty infiltration, common in RC tear muscles [26] and indicative of impaired regeneration [27], has been previously noted in this RC tear cohort via proteomic analysis, thus suggesting a reduction in regenerative capacity [9].

By analyzing the activation and inhibition of myogenic factors in relation to lesion age and patient age, we identified that activation occurs in younger patients and within the first month after acute trauma, while signs of degeneration increase with advancing patient age and lesion age.

Increased densities of myogenic factors in supraspinatus compared to deltoid in younger patients and early trauma phases suggest retained myogenic potency. However, as lesion age increases, particularly for MyoD and myogenin, myogenic activity decreases with patient age. The differential trends in Pax7 (early-stage myogenesis) versus MyoD and myogenin (later-stage markers) highlight a temporal discrepancy: Pax7 density increases with patient age and remains higher than in the deltoid even in older lesions, whereas myoD and myogenin decrease. This could indicate that myogenic cells are activated, but their maturation is subsequently impaired

The observed increase in satellite cell density is not a general reaction pattern. Previous studies have demonstrated a decline in satellite cell numbers with increasing patient age [28,29,30] and as a result of disuse [31]. However, we have also previously reported relatively high numbers of Pax7-positive cells in inclusion body myositis, a chronic condition with limited regenerative capacity [32].

The myogenic repair response to a single lesion generally lasts about two weeks [33,34]. In animal models of tenotomy [35], protein expression related to muscle regeneration can be seen for over two months. Our findings of prolonged myogenic factor expression following supraspinatus tendon tear align with these observations. Our previous proteomic studies indicated upregulation of inflammation, lipid metabolism, and extracellular matrix, all suggesting negative effects on regeneration [9]. Thus, it could be proposed that there is a balance between the positive and negative effects on myogenic cells, which shifts after the initial months after tendon damage.

In conclusion, it appears that satellite cells in the supraspinatus muscles in RC tears are capable of initiating myogenesis, but with increasing patient age and lesion age, many cells become arrested in the initial phase, and their progression to mature muscle is hindered. 

Clinically, this suggests that younger patients may have better muscle recovery outcomes and might benefit more from surgical reconstruction, and early intervention is advantageous. This aligns with a recent long-term study comparing surgical and conservative treatment [36].

Addressing muscle degeneration is critical to treating RC tears, and understanding the cascade of regenerative events is essential for developing effective therapeutic strategies. The fact that myogenesis seems suppressed while the myogenic stem cells are preserved suggests that therapies aimed at recruiting resident stem cells could be beneficial as an adjunct to surgery, as may stem cell therapy.

A limitation of this study is the relatively small and heterogeneous patient cohort. Additionally, we assumed the deltoid muscle was healthy for comparison, though it may be asymptomatically affected [37] or compensatorily loaded, as suggested by the elevated deltoid Pax7 density in early lesions.

Another limitation is the limited tools available for analyzing non-monotonous functions, which complicates result interpretation.

A strength of this study is the use of digital automated quantitation, allowing for in situ analysis of non-dissociated tissue. Another strength is the temporal aspect of the study, which considers both patient age and lesion age.

## 4. Materials and Methods

### 4.1. Patient Cohort

Patients were recruited from the trauma clinic at Odense University Hospital, where they were referred due to a relevant shoulder trauma. Inclusion criteria were a recent magnetic resonance imaging confirming an RC tear involving the supraspinatus tendon and a willingness to undergo surgery. Tears were classified geometrically [38]. Exclusion criteria were severe retraction (more than 2 × 2 cm) of tendons, fatty infiltration higher than Goutallier grade 2 [26,39], diabetes, autoimmune diseases, previous shoulder surgery, fractures, or a dislocated shoulder.

All patients gave informed written consent. The local research ethics committee (The Regional Committees on Health Research Ethics for Southern Denmark, J. No. S-20160037) granted ethical approval, and the study was reported to the Danish Data Protection Agency.

While performing arthroscopic RC tendon repair, a 3 × 3 mm biopsy punch was used to take biopsies from the tendon-near supraspinatus muscle and biopsies from the ipsilateral deltoid muscle present in the operation field for comparison. The biopsies were obtained through the lateral portal under direct visualization from the arthroscope.

Forty-two (42) consecutive patients (age 34–73 years) with RC tendon tears, all involving the supraspinatus, agreed to participate in this study. The age of the lesions was registered from hospital records and as reported by the patients. Three patients were excluded due to insufficient quality of one or both muscle biopsies. Two patients were excluded because the biopsies included the myotendinous junction. Two patients were excluded because the ages of their tendon lesions were unknown. One patient was excluded because a primary muscle disease was suspected, and one because of suprascapular nerve palsy. This resulted in a cohort of thirty-three patients (9 women and 24 men), with a mean age of 59.9 years (Table 3).

### 4.2. Histological Analyses

Biopsies from the supraspinatus muscle were taken approximately 1 cm medial to the tendon under direct visualization from the arthroscope using a 3 × 3 mm biopsy punch through the lateral portal during arthroscopic RC tendon repair. In addition, and for comparison, biopsies from the deltoid muscles were taken as well.

Biopsies were fixed in 10% phosphate-buffered formaldehyde, embedded in paraffin, and cut into parallel 2-µm-thick microtome sections. For immunohistochemically staining, tissue sections were dewaxed in xylene and rehydrated in ethanol. Endogenous peroxidase activity was quenched using 1.5% hydrogen peroxide in tris-buffered saline. Heat-induced epitope retrieval was performed using T-EG buffer (10 mM Tris, 0.5 mM EGTA, pH 9).

Immunohistochemistry for Pax7 and MyoD was performed on a Benchmark Ultra immunostainer (Ventana Medical Systems, Oro Valley, AZ, USA) using mouse anti-Pax7 (1:100, Myeloma strain P3U1, Developmental Studies Hybridoma Bank, University of Iowa, Iowa City, IA, USA), monoclonal rabbit anti-MyoD (1:25, clone EP212, Cell Marque, Rocklin, CA, USA) antibodies, and the OptiView-DAB detection system including counterstain with copper-enhanced hematoxylin. Immunohistochemistry for myogenin was performed on the OMNIS platform (Dako/Agilent, Glostrup, Denmark) using monoclonal rabbit anti-myogenin (1:100, clone EP162, Cell Marque, Rocklin, CA, USA) antibody and EnVisionTM FLEX as detection system including hematoxylin as a nuclear counterstain.

An increased number of internal nuclei is an indicator of regeneration. The number was considered increased when more than 3% of the fibers contained internal nuclei [40].

### 4.3. Estimation of the Number of Pax7, MyoD, and Myogenin-Positive Nuclei per Area

The biopsies were analyzed with Qupath [24]-automated analysis by two independent raters. In the scanned slides, the entire area of muscle fiber tissue was selected with the wand tool, and the total number of Pax7^+^, MyoD^+^, and myogenin^+^ nuclei in the selected area was found by the positive cell detection function, as shown in Appendix B, Figure A2. In each slide, a small region was chosen initially to optimize detection parameters before applying them to the whole area of muscle fibers. The results were calculated as positive nuclei per square mm (density). For Pax7 and MyoD, this is identical to the number of positive cells, while myogenin-positive nuclei are mainly found in multinucleate muscle fibers. Positive and negative controls can be seen in Appendix B, Figure A3.

### 4.4. Statistical Analysis

Patients were divided into equal-sized groups according to lesion age ≤3 months and >3 months. This grouping is also justified due to the latest national Danish clinical guideline recommending treatment of rotator cuff lesions within 3 months after a trauma [41]. Associations between quantification of myogenic factors and the age of the patient (numerical in years) and the age of the lesion were investigated by mixed-effects linear regression, both in univariate and in mutually adjusted models, including random intercepts for each patient and each rater and, separately, for deltoid and supraspinatus. We adjust for age (patient and lesion) as part of the regression models. We did not use any pre-specified coefficient, but fitted the age adjustment as part of the model.

A lowess (locally weighted scatterplot smoothing) curve with a bandwidth of 0.9 was used for the graphic presentation of associations between outcomes and age of lesion and age. Furthermore, we investigated possible differences in age and lesion age by gender using the Wilcoxon rank sum test and Fisher’s exact test.

The presence of internal nuclei was tested by chi-square.

## 5. Conclusions

This study confirmed that an RC tendon lesion induces changes in the affected supraspinatus muscle compared to a control muscle. Our quantitation of myogenic factors indicated that the tendon lesion induces activation of the myogenic response, but also that there is a decrease in myogenic repair capacity, a capacity difference that increases with both patient and lesion age. A clinical conclusion on these observations is that younger patients may have better chances of muscle recovery and may thus benefit more from surgical reconstruction, and that early intervention is favorable. The results of RC surgery in elderly patients may be improved by interventions that support supraspinatus muscle regeneration.

## Figures and Tables

**Figure 1 ijms-25-11742-f001:**
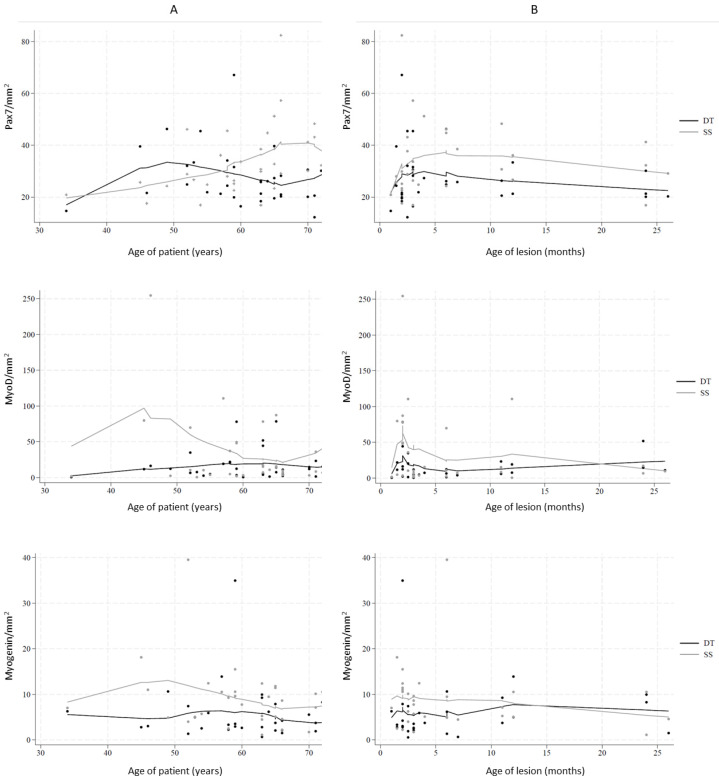
Markers of myogenic regeneration Pax7, MyoD, and myogenin in deltoid muscle (DT) and supraspinatus muscle (SS). (**A**) Number of positive cells per mm^2^ in relation to the age of patients. (**B**) Number of positive cells per mm^2^ in relation to the age of the lesion.

**Table 1 ijms-25-11742-t001:** MyoD, myogenin, and Pax7-positive nuclei in supraspinatus muscle compared to deltoid muscle.

	Adjusted Coefstr	Adjusted *p* Value
Pax7	4.29 (−0.15; 8.72)	0.058
MyoD	47.17(29.95; 64.40)	<0.001
Myogenin	17.80 (5.88; 29.73)	0.003

**Table 2 ijms-25-11742-t002:** Density of MyoD, myogenin (MyoG), and Pax7-positive nuclei related to the age of patients and to lesion age of more than 3 months compared to 3 months or less.

	Muscle	Variable	Adjusted Coefstr	Adjusted *p* Value
Pax7	deltoid	Age of patient	−0.12 (−0.54; 0.31)	0.585
Pax7	deltoid	Age of lesion > 3 months	35.60 (10.13; 61.07)	0.006
Pax7	supraspinatus	Age of patient	0.60 (0.16; 1.04)	0.007
Pax7	supraspinatus	Age of lesion > 3 months	−4.36 (−30.68; 21.96)	0.746
MyoD	deltoid	Age of patient	0.31 (−0.45; 1.08)	0.418
MyoD	deltoid	Age of lesion > 3 months	2.01 (−43.69; 47.71)	0.931
MyoD	supraspinatus	Age of patient	−1.04 (−3.04; 0.96)	0.309
MyoD	supraspinatus	Age of lesion > 3 months	108.09 (−12.19; 228.37)	0.078
MyoG	deltoid	Age of patient	−0.08 (−0.37; 0.21)	0.595
MyoG	deltoid	Age of lesion > 3 months	10.24 (−7.69; 28.17)	0.263
MyoG	supraspinatus	Age of patient	−0.32 (−0.65; 0.02)	0.064
MyoG	supraspinatus	Age of lesion > 3 months	27.87 (7.54; 48.20)	0.007

**Table 3 ijms-25-11742-t003:** Study cohort.

	Age of Lesion up to Three Months				Age of Lesion over Three Months		
#	Age of Patient	Age of Lesion	Gender	#	Age of Patient	Age of Lesion	Gender
	(years)	(months)			(years)	(months)	
1	45	2	f	17	34	4	m
2	46	2	m	18	49	6	m
3	52	3	f	19	52	6	m
4	54	3	m	20	53	12	m
5	58	2	m	21	55	3.5	m
6	59	2	f	22	57	12	m
7	59	2	m	23	58	72	m
8	59	3	f	24	63	7	f
9	60	3	m	25	63	11	m
10	63	2	m	26	63	24	m
11	65	2	m	27	64	6	f
12	66	2	m	28	65	4	m
13	66	3	m	29	65	36	f
14	70	3	m	30	66	26	m
15	71	3	m	31	70	24	f
16	73	3	m	32	71	11	m
				33	72	24	f
Mean	60	2			60	17	
Std.dev.	8.3	0.5			9.5	17.2	

## Data Availability

The data presented in this study are available on request from the corresponding author. The data are not publicly available due to the size of files of scanned whole slides.

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
