# Peer review of "Supraspinatus Muscle Regeneration Following Rotator Cuff Tear: A Study of the Biomarkers Pax7, MyoD, and Myogenin"

_ijms, 2024, doi:10.3390/ijms252111742_

Round 1
Reviewer 1 Report
Comments and Suggestions for Authors
Thank you for the interesting study.
I have the following remarks and requests.
L98 I assume Fig 3 shouls read Fig 1
L106 The continuous increase of Pax7 is not that obvious. A would suggest en overal increase is more appropriate
L120 There seems to be a mix-up of the > and < 3 months. Please correct
L143 Ref 24 should be 27
L159 Ref 25-27 should read 28-30. Please correct the numbering starting from this point
L226 [ ] is missing for ref 24
L228 [ ] missing
Author Response
Thank you for the interesting study.
I have the following remarks and requests.
L98 I assume Fig 3 should read Fig 1
Agree. We have changed the figure number (L99)
L106 The continuous increase of Pax7 is not that obvious. A would suggest en overal increase is more appropriate.
Thank you for this comment. We agree and have changed it accordingly (L107)
L120 There seems to be a mix-up of the > and < 3 months. Please correct
Thank you for pointing this out. We have changed it (L121)
L143 Ref 24 should be 27
L159 Ref 25-27 should read 28-30. Please correct the numbering starting from this point
L226 [ ] is missing for ref 24
L228 [ ] missing
Thank you for pointing out these mistakes. We have been through all references, so that they now should be correct.
Reviewer 2 Report
Comments and Suggestions for Authors
This study analyzes the regenerative potential of the supraspinatus muscle in rotator cuff tear conditions based on the quantification of Pax7, MyoD and myogenin expression. In my opinion, the results are interesting as they show that younger patients with rotator cuff tear maintain a myogenic potential that would allow better chances of muscle recovery by early surgical reconstruction.
Comments:
The authors interpret the presence of + nuclei for Pax7, MyoD1 and myogenin as “regenerative potential” (p. 5, line 137) or “regenerative capacity” (p. 5, line 141). This is correct as it could indicate the start of the regenerative myogenic process. However, it would be interesting if the authors could report whether they also saw regenerative muscle fibers (fibers with central nuclei) as they are also a histological indicator of regeneration (resulting from previous necrosis and inflammation). Although the present study focused on three myogenic markers using immunohistochemistry, a simple hematoxylin-eosin would be sufficient to observe them. If they are not observed, and there is fibroadipose tissue in the biopsy, it should be interpreted as a failure of the regenerative response. The persistence of the labeling in the samples, and the absence of other regenerative indicators, would reinforce the idea that the response “is only initial.” This reinforces, in my opinion, the authors’ conclusion.
On the other hand, it should be considered that the expression of myogenic markers in apparently normal muscle fibers (at least that is the feeling that the images in Fig. B1 convey to me) is not the result of the regeneration of a degenerative process and necrosis of the muscle fibers, but rather of the response of these to tendon injury. Classical tenotomy studies have shown how muscle fibers respond to tendon injury. They suffer not only atrophy, but shortening that implies an adaptive internal remodeling (especially of the myofibrillar apparatus), without the muscle fibers suffering necrosis. This may justify the activation of the satellite cell population and, consequently, the activation of the myogenic program without a process of degeneration and regeneration of muscle fibers occurring. That is, the possibility should be considered that the positivity of myogenic markers may be more related to an adaptive response of internal remodeling than regeneration.
Figures:
Fig. 1. In the graphs in column B, on the axis where it says “Leason” it should say “Lesion”
Discussion:
I suggest that the authors revise the wording of the Discussion. In some paragraphs, I see it disorganized, poorly composed or disjointed. For example:
Page 5, lines (156-157) the sentence should be joined to the following paragraph.
Page 6, lines (219-220). In my opinion the last sentence “A strength of this study is the introduction of the digital automated quantitation design, and it taking into account both patient and injury age” should not be at the end. I think it should be, in the same paragraph, together with the sentence from lines 156-157 and the following paragraph (lines 158-163).
References:
The References section should be thoroughly reviewed:
- Reference 10 is incomplete.
- Many of the references do not include the abbreviated title of the journal. Others do.
- Others are not adequately referenced. For example, references 24 and 28: “Arthroscopy : the journal of arthroscopic & related surgery : official publication of the Arthroscopy Association of North America and the International Arthroscopy Association” ¿?, “The journal of histochemistry and cytochemistry : official journal of the Histochemistry Society” ¿?
Author Response
This study analyzes the regenerative potential of the supraspinatus muscle in rotator cuff tear conditions based on the quantification of Pax7, MyoD and myogenin expression. In my opinion, the results are interesting as they show that younger patients with rotator cuff tear maintain a myogenic potential that would allow better chances of muscle recovery by early surgical reconstruction.
Comments:
The authors interpret the presence of + nuclei for Pax7, MyoD1 and myogenin as “regenerative potential” (p. 5, line 137) or “regenerative capacity” (p. 5, line 141). This is correct as it could indicate the start of the regenerative myogenic process. However, it would be interesting if the authors could report whether they also saw regenerative muscle fibers (fibers with central nuclei) as they are also a histological indicator of regeneration (resulting from previous necrosis and inflammation). Although the present study focused on three myogenic markers using immunohistochemistry, a simple hematoxylin-eosin would be sufficient to observe them. If they are not observed, and there is fibroadipose tissue in the biopsy, it should be interpreted as a failure of the regenerative response. The persistence of the labeling in the samples, and the absence of other regenerative indicators, would reinforce the idea that the response “is only initial.” This reinforces, in my opinion, the authors’ conclusion.
On the other hand, it should be considered that the expression of myogenic markers in apparently normal muscle fibers (at least that is the feeling that the images in Fig. B1 convey to me) is not the result of the regeneration of a degenerative process and necrosis of the muscle fibers, but rather of the response of these to tendon injury. Classical tenotomy studies have shown how muscle fibers respond to tendon injury. They suffer not only atrophy, but shortening that implies an adaptive internal remodeling (especially of the myofibrillar apparatus), without the muscle fibers suffering necrosis. This may justify the activation of the satellite cell population and, consequently, the activation of the myogenic program without a process of degeneration and regeneration of muscle fibers occurring. That is, the possibility should be considered that the positivity of myogenic markers may be more related to an adaptive response of internal remodeling than regeneration.
Thank you very much for pointing this out. We agree with suggestion that looking at the nuclear positions could contribute to the study. Increased number of internal nuclei was present in 18 of the 33 supraspinatus biopsies and we think this supports the view of activation of the myogenic cascade. We have now included a registration of patients with increased number of internal nuclei and correlated the presence with patient and lesion age (L136, L160,L270 and Appendix C).
Figures:
Fig. 1. In the graphs in column B, on the axis where it says “Leason” it should say “Lesion”
Thank you for pointing this out. We have changed it.
Discussion:
I suggest that the authors revise the wording of the Discussion. In some paragraphs, I see it disorganized, poorly composed or disjointed. For example:
Page 5, lines (156-157) the sentence should be joined to the following paragraph.
Page 6, lines (219-220). In my opinion the last sentence “A strength of this study is the introduction of the digital automated quantitation design, and it taking into account both patient and injury age” should not be at the end. I think it should be, in the same paragraph, together with the sentence from lines 156-157 and the following paragraph (lines 158-163).
Thank you for this comment. In response, we have rewritten the discussion and hope that it now appears more organized. In doing this, we also have moved the last sentence to the paragraph on quantification.
References:
The References section should be thoroughly reviewed:
We apologize for the disorder in the references that arose when the position in the manuscript of materials and methods were changed. We have gone through the references to correct this.
- Reference 10 is incomplete.
Thank you for pointing this out. We have corrected it.
- Many of the references do not include the abbreviated title of the journal. Others do.
Thank you for pointing this out. We have changed the style of abbreviation in Endnote.
- Others are not adequately referenced. For example, references 24 and 28: “Arthroscopy : the journal of arthroscopic & related surgery : official publication of the Arthroscopy Association of North America and the International Arthroscopy Association” ¿?, “The journal of histochemistry and cytochemistry : official journal of the Histochemistry Society” ¿?
We agree. We have changed the output style in Endnote.
Reviewer 3 Report
Comments and Suggestions for Authors
Kildall Hejbol et al. analysed markers of intrinsic muscle regeneration (Pax7, MyoD and myogenin) in 2 muscles of 33 patients following a rotor cuff injury.
Abstract lacks detail with respect to patient population (age range, gender) and lesion age.
Results / discussion: Variation between measurements of 2 raters should be explained in more details in results (e.g. how many fields of view were analysed, variation between fields, systemic variation?) and discussion should be more elaborate. Authors suggest systemic variation due to e.g. thresholding differences, but figure 1a doesn’t show a consistent pattern of rater 2 e.g. always being higher. Line 219: “A strength of this study is the introduction of the digital automated quantification design”. How is this a strength given the inter rater variability?
Data is presented /mm2. Large variation in number of MyoD/Pax/Myogenin positive nuclei/mm2 is seen between patients, but this is not being discussed. Is this the result of more nuclei or a higher proportion of nuclei that is positive for analysed marker? It would be informative if data would also be presented as percentage of nuclei positive.
Hypothesis is that hampered muscle regeneration capacity would be reflected in the expression of myogenic regulating factors. It is unclear to me how this is addressed in the study. Which patients had hampered muscle regeneration and how was this assessed?
Line 188. Upon muscle damage, how long it is expected that these makers are increased? How long does the healing process generally take and how does this relate to "lesion age"?
Why was age of lesion less or more than 3 months chosen (table 2)?
Pax7 and MyoD appear to increase in first 2-3 months after lesion in both DT and SS muscle, this needs to be discussed.
#31 analyses
#31 rephrase to improve clarity
#155 study of Koide, comparable with respect to time of analysis to your study?
Methods:
Which biopsy method/tool was used?
Table 3: add descriptive statistics (mean +/-sd for age, lesion) per group
How were nuclei stained, haematoxylin?
Figure A1, given the large variation in number of + nuclei/mm2 it is difficult to compare 2 raters. A splitted y-axis would be helpful.
Comments on the Quality of English LanguageEnglish in generally fine, but some sentences could be improved by a native English speaker
Author Response
Kildall Hejbol et al. analysed markers of intrinsic muscle regeneration (Pax7, MyoD and myogenin) in 2 muscles of 33 patients following a rotor cuff injury.
Abstract lacks detail with respect to patient population (age range, gender) and lesion age.
Thank you for pointing this out. We have added details in the abstract (L25)
Results / discussion: Variation between measurements of 2 raters should be explained in more details in results (e.g. how many fields of view were analysed, variation between fields, systemic variation?) and discussion should be more elaborate. Authors suggest systemic variation due to e.g. thresholding differences, but figure 1a doesn’t show a consistent pattern of rater 2 e.g. always being higher. Line 219: “A strength of this study is the introduction of the digital automated quantification design”. How is this a strength given the inter rater variability?
Thank you for this comment. The use of the automated counting system allow for selection of the entire skeletal muscle area in a section and subsequent estimation of stained nuclei in the entire area as described in material and methods (L 274) and shown in the figure in appendix B. Thus, only one field was created for each section. We think that the variation is due to differences in the set of threshold modified by how connective tissue, artifacts, and empty areas are excluded. The set of threshold is in part personal but could also be influenced by differences in color tone of the screens used for the analyses. We agree that the value of including two observers can be debated, and we have deleted the statement.
Data is presented /mm2. Large variation in number of MyoD/Pax/Myogenin positive nuclei/mm2 is seen between patients, but this is not being discussed. Is this the result of more nuclei or a higher proportion of nuclei that is positive for analysed marker? It would be informative if data would also be presented as percentage of nuclei positive.
Thank you for pointing this out. Positive cells in percentage of total cells is a common and informative way of normalizing quantitative data. However, in a study in progress, we have seen that there are problems with quantification of haematoxylin stained nuclei in immunohistochemically stained sections. Thus in neighbour sections stained on automated platforms with different protocols, like in this study, there can be a more than 10% difference in registered nuclei even in large sections. Most likely, this is a consequence of the haematoxylin stain combined with tissue fixation and heat retrieval, Knowing this, we decided not to use nucleus counts for normalization.
Hypothesis is that hampered muscle regeneration capacity would be reflected in the expression of myogenic regulating factors. It is unclear to me how this is addressed in the study. Which patients had hampered muscle regeneration and how was this assessed?
Thank you for pointing this out. Our analysis of this point is based on the argument that activated satellite cells will proliferate and thus increase in number. Subsequently half of them will progress in the myogenic program, loose Pax7 expression and express first MyoD then myogenin. The other half will return to be quiescent Pax7 positive satellite cells. MyoD and myogenin should therefore stay elevated as long as Pax7 is elevated. However, what we see in the Lowess curves is that Pax7 increases while the other factors drop. This is taken as an indication that the progression in the myogenic program from the Pax7 positive stage to subsequent stages is hampered.
Line 188. Upon muscle damage, how long it is expected that these makers are increased? How long does the healing process generally take and how does this relate to "lesion age"?
Pax7 and MyoD appear to increase in first 2-3 months after lesion in both DT and SS muscle, this needs to be discussed.
Thank you for pointing this out. The myogenic program takes about two weeks. However, in muscle disorders the stem cells can be activatd over a longer period. Tenopathy literature have reported presence of protiens indicating active regeneration after more than 2 month post tenotomy. We have now addressed that in the discussion (L192)
Why was age of lesion less or more than 3 months chosen (table 2)?
Thank you for the question. We initially chose this division as our national guide lines recommend operation before 3 months (L283) and because three months divide the patients in equal groups. The Lowess analyses supported the choice (L116).
#31 analyses
#31 rephrase to improve clarity
We agree and have changed the sentence (L33).
#155 study of Koide, comparable with respect to time of analysis to your study?
Thank you for this question. Koide et al states the patient ages, which are similar to those in our patient population. This is now mentioned (L162). There is in Koide´s paper no information on lesion age.
Methods:
Which biopsy method/tool was used?
Thank you for pointing this out. The biopsies were taken using a 3 × 3 mm biopsy punch. This now appears in Methods (L251).
Table 3: add descriptive statistics (mean +/-sd for age, lesion) per group .
Thank you for pointing this out. Information om mean +/-sd for age and lesion have been added to Table 3 (L248).
How were nuclei stained, haematoxylin?
Thank you for pointing that out. The counter stain used in Pax7 and MyoD stains is hematoxylin with copper enhancement as part of the OptiView kit provided by Ventana, and in myogenin stain hematoxylin as part of the Envision kit provided by DAKO. This is now included in the Methods (L 265 and L269).
Figure A1, given the large variation in number of + nuclei/mm2 it is difficult to compare 2 raters. A splitted y-axis would be helpful.
We think this is an excellent suggestion and have redrawn Figure A1.
Comments on the Quality of English Language
English in generally fine, but some sentences could be improved by a native English speaker
We have addressed this consulting a skilled English speaker.
Round 2
Reviewer 1 Report
Comments and Suggestions for Authors
Thank you for the extensive revision
Reviewer 3 Report
Comments and Suggestions for Authors
Manuscript has been substantially improved